# Analysis of Land Cover Change Detection in Gozamin District, Ethiopia: From Remote Sensing and DPSIR Perspectives

**Abebaw Andarge Gedefaw [1,2,\*]** , **Clement Atzberger [1]** , **Thomas Bauer [1]** ,
**Sayeh Kassaw Agegnehu [2] and Reinfried Mansberger [1]**

[1] Institute of Geomatics, University of Natural Resources and Life Sciences Vienna, Peter-Jordan-Strasse 82, 1190 Vienna, Austria; clement.atzberger@boku.ac.at (C.A.); t.bauer@boku.ac.at (T.B.); mansberger@boku.ac.at (R.M.)

[2] Institute of Land Administration, Debre Markos University (DMU), 269 Debre Markos, Ethiopia; sayehalem@gmail.com

\* Correspondence: abebaw.gedefaw@students.boku.ac.at

**Abstract:** Land cover patterns in sub-Saharan Africa are rapidly changing. This study aims to quantify the land cover change and to identify its major determinants by using the Drivers, Pressures, State, Impact, Responses (DPSIR) framework in the Ethiopian Gozamin District over a period of 32 years (1986 to 2018). Satellite images of Landsat 5 (1986), Landsat 7 (2003), and Sentinel-2 (2018) and a supervised image classification methodology were used to assess the dynamics of land cover change. Land cover maps of the three dates, focus group discussions (FGDs), interviews, and farmers' lived experiences through a household survey were applied to identify the factors for changes based on the DPSIR framework. Results of the investigations revealed that during the last three decades the study area has undergone an extensive land cover change, primarily a shift from cropland and grassland into forests and built-up areas. Thus, quantitative land cover change detection between 1986 and 2018 revealed that cropland, grassland, and bare areas declined by 10.53%, 5.7%, and 2.49%. Forest, built-up, shrub/scattered vegetation, and water bodies expanded by 13.47%, 4.02%, 0.98%, and 0.25%. Household surveys and focus group discussions (FGDs) identified the population growth, the rural land tenure system, the overuse of land, the climate change, and the scarcity of grazing land as drivers of these land cover changes. Major impacts were rural to urban migration, population size change, scarcity of land, and decline in land productivity. The outputs from this study could be used to assure sustainability in resource utilization, proper land use planning, and proper decision-making by the concerned government authorities.

**Keywords:** land cover change; remote sensing; image analysis; supervised classification; maximum likelihood; DPSIR

## 1. Introduction

Information about land cover is required for planning issues and sustainable management of natural resources, as land cover has substantial impacts on the functioning of socio-economic and environmental systems, with significant tradeoffs for sustainability, biodiversity socioeconomic vulnerability of people and ecosystems [1–4]. Reid et al. [5] stated that land cover change is enhanced by human activities and natural processes. The change is a result of complex interactions between social, economic, and biophysical conditions, which may occur at various temporal and spatial scales, and are often caused by agricultural diversifications, advancements in technology, and population pressure [5]. Knowledge about land cover is mandatory for land use planning and policy development.

It is also a precondition for the monitoring of land use, for the modeling of environmental changes, and for receiving land use statistics at all levels of administration [5]. Recently, studies to detect changes in land cover have attracted the attention of numerous researchers [6,7]. It is also an issue that requires investigation into sustainable land management [8]. On the other side, it exerts influence on biodiversity, the hydrological cycle, land productivity, and sustainable natural environment [9].

Remote sensing data are proper sources for assessing land cover [10–14]. With the invention of remote sensing techniques, land cover mapping has given a useful and detailed way to improve the selection of areas designed to agricultural and urban areas of a region [15]. Remote sensing technology is also important for monitoring and quantifying the natural resources and dynamic phenomena on the Earth's surface [16]. In recent years, remote sensing data have effectively assessed long-term changes in vegetation cover [17]. Satellite imagery is a cost-effective tool to capture and analyze land cover data over large geographic regions. In the last decades, several techniques of land cover mapping and change detection have been developed and applied [18–26].

Rapid land cover changes are observed globally [27]. Thus, land cover change detection and analysis are very crucial for understanding landscape dynamics over a known time frame [28]. Population growth and economic activities have quickly transformed land cover [29]. Humans and the interaction between natural and anthropogenic processes have significantly changed the surface of the Earth through time [28,30–34]. Other studies confirmed that anthropogenic processes (e.g., land fragmentation) are changing the land cover on spatial and temporal scales with effects on the whole ecosystem [35–42]. Previous studies indicated different views regarding the main drivers for land cover change [12,28,43,44]. Mather and Needle (2000) stated population growth as the key factor for land cover change and the often resulting land degradation [44]. Studies of Wang et al. (2008) indicated socioeconomic development as the main driving force of land cover change in the Tibetan plateau (China) [45]. Others proved in their investigations that land cover change is a combination of the effects of anthropogenic activities, such as expansion of farm land, and fundamental social processes, such as population growth, together with impacts of policy, institutional settings, and cultural factors [46–49]. Lambin et al. (2003) found that neither population nor poverty alone constitute major underlying causes of land cover change at the global scale. They concluded that the main driver of land cover change are people's responses to economic opportunities facilitated by institutional factors [12]. In the northern highlands of Ethiopia, other studies concluded that intensive permanent agriculture, deforestation, and encroachment of cultivation were the major causes of land cover change and land degradation. Recently, a large amount of observed land cover change was attributed to socioeconomic and biophysical drivers, such as population growth, agricultural expansion and intensification, accessibility to infrastructure, climate, and invasive alien plant species which affect the ecosystem [50,51]. Studies conducted in Ethiopia revealed that population pressure, deforestation, agricultural expansion, and lack of alternative livelihoods, government land policy, investment, overgrazing, and land tenure system are the main drivers for the land cover change [43,49,52].

Currently, the total population of Ethiopia is about 113 million people. The growth rate is 2.61%, which is an increase of approximately 2.2 million people per year. According to forecasts, its population will be more than doubled by the year 2050 [53]. This population growth causes an increasing pressure on land resources, primarily resulting from an increasing demand for agricultural products [43,52,54]. Population pressures in Ethiopia have already decreased the size of holdings, both for arable and grasslands, leading to alteration of forested and peripheral areas into agrarian lands [55]. High population pressure often leads to deforestation, overgrazing, land degradation and decreasing agricultural productivity in the upstream parts of the Blue Nile basin [56].

This study analyzes the magnitude, extent, and rate of spatio-temporal land cover change over the time period for the last 32 years in the Gozamin District of Ethiopia. Detailed spatial maps on land cover and land cover changes are gained by using multi-spectral satellite imagery from Landsat (1986, 2003) and Sentinel-2 (2018). The interacting components of social, economic, and environmental systems with land cover change are assessed using the DPSIR (Drivers, Pressure, State, Impact, Response)

framework. DPSIR serves to identify the main factors for land cover changes in the study area and to compare farm households' perception of land cover changes with land cover changes assessed from remote sensing data.

## 2. Materials and Methods

### 2.1. Study Area

The study was carried out in Gozamin district (see Figure 1), located 270 km east of the regional capital Bahir Dar, Amhara National Regional State, and 300 km northwest from Addis Ababa, Ethiopia. The district has a total area of 1218 km$^2$ and contains in total 25 rural kebeles (lowest administrative level in Ethiopia). Its elevation ranges from ~1000 meters to ~3200 meters above sea level. Annual rainfall ranges from about 1000 mm to 1510 mm per year. The average minimum temperature is 8.5 °C and the maximum temperature reaches 30 °C. The district is characterized by a relatively flat landscape, flood plains, and wetlands [57]. Gozamin district has 134,000 inhabitants with almost gender balance. The population density is 109 people per km$^2$. Sedentary, rainfed, small-scale agriculture, and livestock rearing are the major sources of livelihood within the Gozamin district. People primarily perform mixed cereal agriculture, with farmers growing teff, finger millet, sorghum, maize, barley, wheat, pulses, oil crops, vegetables, and fruits [58].

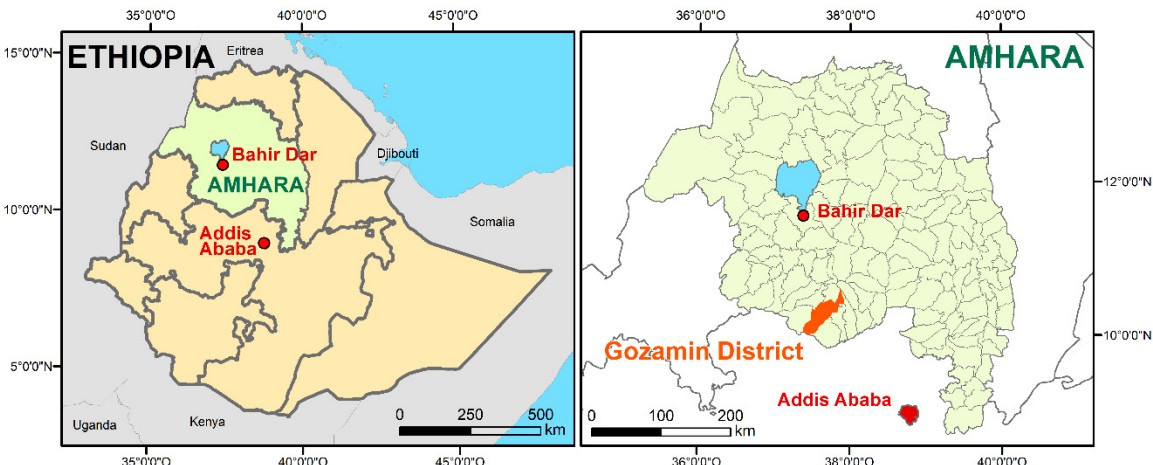

**Figure 1.** Gozamin district—study area.

### 2.2. Methods

#### 2.2.1. Remote Sensing Data and Preprocessing

Satellite images of Landsat 5 TM from 1986, Landsat 7 ETM+ from 2003, and Sentinel-2 from 2018 were used to produce the land cover maps. The selection of the years of investigation was according to key events in history with a significant influence on the land cover change: the year 1986 gave evidence of land cover before the government change of 1991. The year 2003 showed the land cover before the execution of the rural land certification program after 2002. Finally, the year 2018 was characterizing the current situation.

The main specifications of the three sensors and the derived datasets are summarized in Table 1. Landsat images with a spatial resolution of 30 m were downloaded from United States Geological Survey (USGS) (https://earthexplorer.usgs.gov/). The atmospherically-corrected Sentinel-2 images were downloaded from the service platform implemented at the University of Natural Resources and Life Science, Vienna (BOKU) [59]. The images were geometrically co-registered, ortho-rectified, and atmospherically corrected. Multi-temporal images assessed by different sensors were resampled to 20 m resolution, applying nearest neighbor resampling because of the ability to preserve the original

values in the unaltered scene [60]. Images in the period from January to February (dry season) were taken for training and classification, as this is the best time to spectrally distinguish vegetation cover and for getting cloud-free satellite images of the surface.

**Table 1.** Description of the satellite images used in this study.

| Satellite | Sensor | Acquisition Date | Bands Used | Spatial Resolution |
|---|---|---|---|---|
| Sentinel-2 | Multispectral Imager (MSI) | 2018/2/6 | Visible (B2, B3, B4)<br>NIR (B8, B8A)<br>Red Edge (B5, B6, B7)<br>SWIR (B11, B12) | 10 m<br>10 and 20 m<br>20 m<br>20 m |
| Landsat 7 | Enhanced Thematic Mapper (ETM+) | 2003/2/4 | Visible (B1, B2, B3)<br>NIR (B4)<br>SWIR (B5, B7) | 30 m<br>30 m<br>30 m |
| Landsat 5 | Thematic Mapper (TM) | 1986/1/12 | Visible (B1, B2, B3)<br>NIR (B4)<br>SWIR (B5, B7) | 30 m<br>30 m<br>30 m |

### 2.2.2. Field Data Collection

Ground reference data were collected for training and validation of each land cover type in the study area. There is no universally accepted single benchmark of sample size for reference data points. According to [61], a minimum of 50 samples for each of land cover classes and fewer than 12 classes should be collected for maps with area covering less than 4000 km$^2$.

To ensure a representative sampling, at least 80 samples for each of the seven land cover classes (Table 2) were selected. The seven classes of land cover were defined according to the Africover classification scheme of Food and Agriculture Organization (FAO) [62]: forest, grassland, cropland, built-up areas, shrubs and scattered vegetation, bare land, and water bodies.

**Table 2.** Description of land cover types identified for analysis in Gozamin district.

| Land Cover Type | Description of Each Land Cover Type |
|---|---|
| Forest | Natural forest and plantation area with mainly planted eucalyptus trees. Areas covered by trees (eucalyptus) forming closed or nearly closed canopies; forest; plantation forest; dense (50%-80% crown cover) predominant species like *Juniperus procera*. |
| Grassland | Land covered with natural grass, or dominated by grass, it includes areas used for communal grazing as well as a bare land that is seasonally grass-covered. |
| Cropland | Rain-fed agricultural land, both small- and large-scale, cropped at least once per year. Areas of land prepared for crop production. This category includes areas currently covered by crops, areas prepared for cultivation, and fallow plots. |
| Built-up areas | Areas are mainly scattered rural settlement and rural institutions such as schools and clinics. |
| Shrubs and scattered vegetation | Land covered by small trees, thorny bushes, and short shrubs, in some cases mixed with grasses, less dense than forests. |
| Bare land | Areas of land which are not covered by any type of vegetation due to erosion, over grazing and cultivation. Areas without any vegetation due to either erosion or mismanagement (especially over grazing); also covered by bare soil and exposed rocks. |
| Water bodies | Permanent rivers and fresh water (rivers, streams, intermittent ponds and canals). It also includes wetlands, which dry up during the dry season. |

The land cover classification was carried out for each acquisition date. Reference data for 1986 and for 2003 were collected by visual interpretation from Google Earth time lapse using pure pixels of 30 m × 30 m for each land cover type. A total of 2054 reference data points were collected for 1986 and 2003, of which 800 were used for classification and 1254 for accuracy assessments. Reference

data for 2018 were collected by field survey between September 2017 and February 2018 using a handheld Global Navigation Satellite System (GNSS) (Garmin GPS MAP 60 CSx). In total, 2275 samples were collected for the land cover classification of 2018, of which 950 were used for training and 1325 assessments for accuracy evaluation. With regard to an existing time lag between the acquisition date of satellite images and the assessment of reference data (Google Earth data and field survey), the reliability of reference data was checked in group discussions and interviews with farmers.

### 2.3. Land Cover Classification and Post Classification

The land cover classifications for the three years (1986, 2003, and 2018) were carried out by supervised pixel-based classification with maximum-likelihood classifier (MLC). This technique was selected as it takes the normal distribution of cloud of points and parameters to compute the statistical probability of a given pixel value being a member of a particular land cover class [61]. In addition to the reflectance values, this tool considers the covariance of the information contained in the sensors' spectral bands of land cover classes [63]. Finally, this approach has a higher probability to consider minority classes that can be swamped by larger classes in unsupervised training.

Supervised classification is based on reference data where land cover is known. For each of the seven land cover types, spectral signatures were derived from the reference data as described before (see Appendix A). Based on these data, a maximum likelihood classification was applied to produce the land cover maps of 1986, 2003, and 2018 for the whole study area.

Post-classification enhancement was carried out with the purpose of increasing classification accuracy and to reduce misclassifications [64]. For this, smoothing algorithms were employed. In such operations a moving window is passed through the classified data set and the majority class within the window is determined. The size of the majority filter window was chosen to be 7 by 7, as larger kernel sizes would produce a higher smoothing in the classification image [61].

### 2.4. Accuracy Assessment

After classification and post-classification enhancement, ground verification was done in order to check the precision of the classified land cover map [65]. The accuracy for the classification of 1986 and of 2003 was determined by the 1254 validation points. The accuracy assessment of the 2018 classification of the Sentinel-2 satellite images was based on the 1325 ground truth validation points collected during the 2017/2018 fieldwork survey (in-situ data). The accuracy of the classification was assessed using randomly selected reference sample points. The accuracy measures, such as overall accuracies, kappa coefficients, user's and producer's accuracies were calculated and an error matrix of the land cover classification was generated [66,67].

### 2.5. Land Cover Change Analysis

Land cover changes were calculated between three different time periods: 1986 to 2003, 2003 to 2018, and 1986 to 2018 using cross-tabulation [68–70]. Percentage changes for each land cover type over time [69,71,72] were calculated. Specific class gains and losses as well as total change and net changes of the study area were assessed [70,73]. In addition, annual change rates were calculated for each land cover type [74,75].

### 2.6. Household Survey and Data Analysis of DPSIR Framework of Land Cover Change

Household surveys were carried out to comprehend the development of the land cover situation and to get insight from landholder household farmers about the DPSIR of land cover change. A two-stage sampling technique was applied to select household farmers. In the first stage, Yebona-Erjena, Adisna-Guilit, and Chimit were selected as representative kebeles (lowest administrative unit) based on a reconnaissance survey between land administration experts, development agents, and kebele administrative officials. In the second stage, 343 household farmers were selected randomly in proportion to the total number of household farmers in each kebele. The sample size was determined

based on the sample size determination equation given by Cochran [76] and was representative for the objective of this investigation due to the relative homogeneity of the household farmers in terms of the cultural, resource endowments, physical, and environmental conditions. The perceptions of household farmers of the drivers, pressures, state, impact, and response of land cover dynamics were collected using a semi-structured questionnaire. The questionnaire included questions to assess general information about households, farmers, and land cover history of the household parcels. In addition, opinions of farmers with regards to the drivers, pressures, state, impact, and response of land cover changes were requested.

The collected data were analyzed through descriptive statistics (percentages and frequencies). For interpretation, the questionnaire results were complemented with qualitative results gained in focus group discussions (FGDs), which were conducted with representatives of local communities from the Gozamin district. In total, three FGDs were conducted, each with six to ten members of local communities. The composition of participants was balanced concerning age and gender. Information accessed in FGD was analyzed through thematic analysis, narrations, and qualitative descriptions.

*2.7. DPSIR Model for Identification of Factors of Land Cover Changes*

To manage land cover in a sustainable way, it is necessary to understand the causes (drivers, pressures) of change and their interactions. To this end, we used the DPSIR framework. The DPSIR framework was developed by the Organization for Economic Co-operation and Development (OECD, 1994) and has been used widely by international agencies [77–79]. This framework helps to understand the interacting factors and interfaces that change the environment. Drivers are forces that cause socio-economic and socio-cultural forces that change in order to fulfill basic needs. These forces can be global, regional, or local. The drivers can be human activities that exert pressure on the environment. Pressures are stressors caused by driving forces on the environment, such as land cover change. State is the condition of the land cover in terms of its constituents. The state of land cover may be altered depending on the pressures exerted. Impacts are changes in land cover that affect human well-being. Responses are the reactions of humans to perceived change of land cover. Responses can be at different levels, including policy and local actions for remediation. Responses can address the pressures or attempt to maintain or improve the state of the land cover [77].

Land cover data from three different acquisition dates (1986, 2003, 2018) were useful to analyze landscape state, to detect land use changes, and to monitor potential management responses. The (DPSIR) model is a proper tool to assess cause-effect relationships between interacting components of social, economic, and environmental systems, like:

- Drivers of land cover change;
- Pressures on the land cover;
- State or condition of land due to the changing situations;
- Impacts on population, economy, ecosystems, and/or environment to the land cover change; and
- Response of the society to the land cover change.

Figure 2 shows how human activity exerts pressure on the land resources and, consequently, changes the state of the environment or land. The state of the environment or land can have influences on people's health, ecosystems, and natural resources. These impacts can result in responses in the form of management approaches, polices, or actions that alter the driving forces, pressures, and, ultimately, the state of the environment. Changes in impacts over time can result in people modifying their response to those impacts [80].

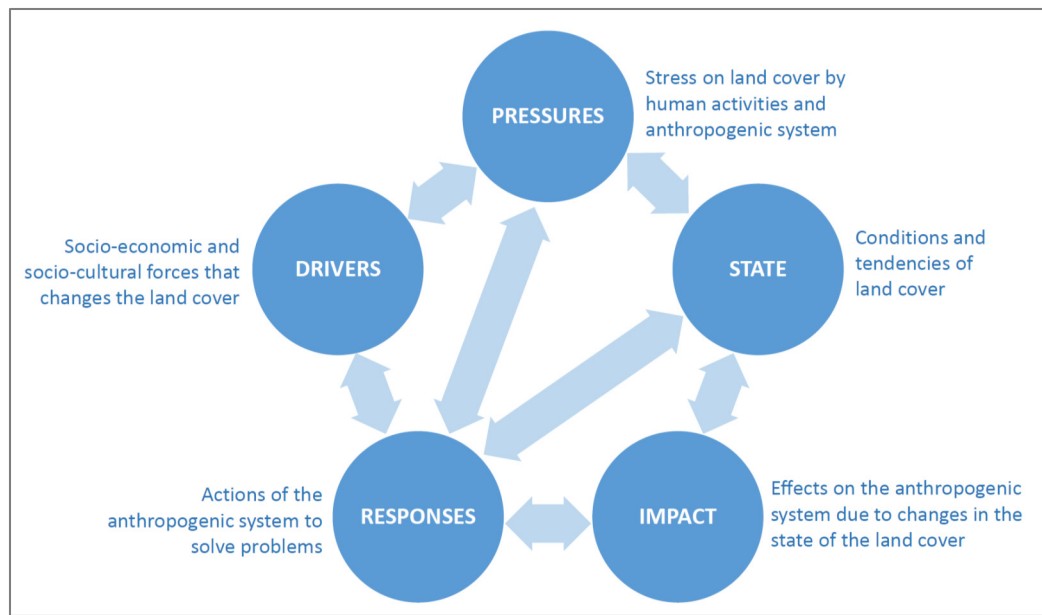

**Figure 2.** The DPSIR (Drivers, Pressure, State, Impact, Response) conceptual framework indicators for land cover changes with their descriptions (adapted in accordance to UNEP, 2007 [77] and European Environment Agency, 2003 [80]).

## 3. Results

### 3.1. Land Cover

The results of land cover classification of Gozamin district for the years 1986, 2003, and 2018 are documented graphically in Figure 3. Quantitative details about the land cover in the respective years are presented in Table 3.

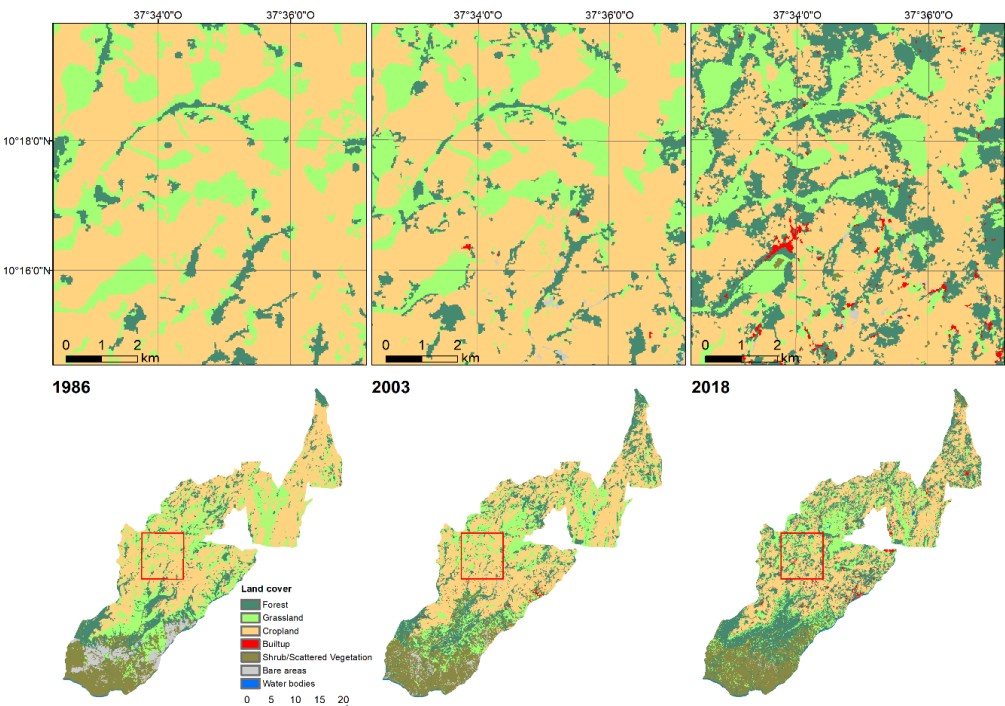

**Figure 3.** Land cover maps of Gozamin district for 1986, 2003, and 2018.

**Table 3.** Total area coverage between the years 1986, 2003, and 2018 for the classified land cover categories.

| Land Cover Type | Area | | | | | |
| --- | --- | --- | --- | --- | --- | --- |
| | 1986 | | 2003 | | 2018 | |
| | (ha) | (%) | (ha) | (%) | (ha) | (%) |
| Forest | 18,630.4 | 15.30 | 22,880.2 | 18.79 | 34,959.2 | 28.77 |
| Grassland | 31,336.7 | 25.72 | 25,301.6 | 20.77 | 24,392.2 | 20.02 |
| Cropland | 52,640.4 | 43.21 | 52,360.1 | 42.98 | 39,813.8 | 32.68 |
| Built-up | 548.8 | 0.45 | 1961.1 | 1.61 | 5451.3 | 4.47 |
| Shrub/Veg | 13,070.3 | 10.73 | 15,536.0 | 12.75 | 14,259.7 | 11.71 |
| Bare areas | 5479.1 | 4.50 | 3630.2 | 2.98 | 2526.6 | 2.01 |
| Water body | 115.1 | 0.09 | 151.6 | 0.12 | 418 | 0.34 |

During the last three decades (1986, 2003, and 2018), the gross changes in area coverage varied from one land cover type to another. Forest and built-up type experienced significant rise, whereas grassland, cropland, and bare areas experienced a reduction in the study area.

*3.2. Accuracy of Land Cover Maps*

The accuracy assessment of the supervised land cover classification is documented in Table 4. It shows an overall accuracy of 87.7% for 1986, 89.2% for 2003, and 94.9% for 2018. The kappa coefficients for 1986, 2003, and 2018 are 0.83, 0.86, and 0.93, respectively. The higher accuracy of Sentinel image (2018) is related to the higher resolution of the images compared to Landsat. The land cover maps fulfilled the accuracy requirements and therefore were used for the further analysis and change detection. In some areas a distinction between spectral similar land cover classes such as bare areas and built-up areas led to lower accuracies. In general, it has to be stated that the analysis of land cover served to identify trends and compare them with the qualitative analysis.

**Table 4.** Accuracy assessment (in %) of land cover maps (1986–2018).

| Land Cover Type | 1986 | | 2003 | | 2018 | |
| --- | --- | --- | --- | --- | --- | --- |
| | User's Accuracy | Producer's Accuracy | User's Accuracy | Producer's Accuracy | User's Accuracy | Producer's Accuracy |
| Forest | 96.5 | 93.9 | 94.9 | 98.9 | 96.5 | 98.8 |
| Grassland | 94.7 | 96.4 | 96.1 | 93.7 | 97.5 | 96.3 |
| Cropland | 79.8 | 98.4 | 80.4 | 97.5 | 96.4 | 91.9 |
| Built-up | 87 | 50 | 96 | 55 | 85.7 | 98.3 |
| Shrub/Sc. Vegetation | 87.5 | 89.7 | 100 | 90.9 | 94.1 | 80 |
| Bare areas | 85 | 63 | 98 | 78 | 95.1 | 75.7 |
| Water bodies | 100 | 75 | 100 | 96.3 | 100 | 97.1 |
| Overall accuracy | 87.7% | | 89.2% | | 94.9% | |
| Kappa statistics | 0.83 | | 0.86 | | 0.93 | |

*3.3. Land Cover Changes*

Table 5 highlights the rate of the changes of land cover in the Gozamin district for the periods from 1986 to 2003, from 2003 to 2018, and it documents the land cover changes during the whole observation period from 1986 to 2018. The results give evidence that the study area was affected by substantial land cover changes.

**Table 5.** Percentage change, net change, and rate of change occurring between the years 1986, 2003, and 2018 for the classified land cover categories of Gozamin district.

| Land Cover Type | Change (%) | | | Net Change (ha) | | | Rate of Change (ha/Year) | | |
|---|---|---|---|---|---|---|---|---|---|
| | 1986–2003 | 2003–2018 | 1986–2018 | 1986–2003 | 2003–2018 | 1986–2018 | 1986–2003 | 2003–2018 | 1986–2018 |
| Forest | 3.49 | 9.98 | 13.47 | 4249.8 | 12,079 | 16,328.8 | 249.9 | 805.3 | 510.3 |
| Grassland | −4.95 | −0.75 | −5.7 | −6035.1 | −909.4 | −6944.5 | −355.0 | −60.6 | −217.0 |
| Cropland | −0.23 | −10.3 | −10.53 | −280.3 | −12,546.3 | −12,826.6 | −16.5 | −836.4 | −400.8 |
| Built-up | 1.16 | 2.86 | 4.02 | 1412.3 | 3490.2 | 4902.5 | 83.1 | 232.7 | 153.2 |
| Shrub/Veg | 2.02 | −1.04 | 0.98 | 2465.7 | −1276.3 | 1189.4 | 145.0 | −85.1 | 37.2 |
| Bare areas | −1.52 | −0.97 | −2.49 | −1848.9 | −1103.6 | −2952.5 | −108.8 | −73.6 | −92.3 |
| Water body | 0.03 | 0.22 | 0.25 | 36.5 | 266.4 | 302.9 | 2.1 | 17.8 | 9.5 |

During the period from 1986 to 2003, the areal coverage of forest, built-up, shrub/scattered vegetation, and water body increased by 4249.8 ha (3.49%), 1412.3 ha (1.16%), 2465.7 ha (2.02%), and 36.5 ha (0.03%), respectively. On the other hand, grassland, cropland, and bare areas decreased by 6035.1 ha (4.95%), 280.3 ha (0.23%), and 1848.9 ha (1.52%), respectively. This was due to the conversion of grassland, cropland, and bare areas to forest, built-up, and shrub/scattered vegetation.

Likewise, forest, with 12,079 ha (9.98%), and built-up areas, with 3490.2 ha (2.86%), showed a significant increment in the period from 2003 to 2018. At the same time, cropland decreased by 12,546.3 ha (10.3%).

The analysis showed that between 1986 and 2003, forest, built-up, shrub/scattered vegetation, and water bodies increased at a rate of 249.9 ha/year, 83.1 ha/year, 145.0 ha/year, and 2.1 ha/year, respectively. Grassland, cropland, and bare areas decreased by 355.0 ha/year, 16.5 ha/year, and 108.8 ha/year, respectively. Similarly, between 2003 and 2018, forest, built-up, and water bodies persistently increased at a rate of 805.3 ha/year, 232.7 ha/year, and 17.8 ha/year, respectively. However, grassland, cropland, and bare areas decreased by 60.6 ha/year, 836.4 ha/year, and 73.6 ha/year, respectively. Unlike the first period, between 2003 and 2018, shrub/scattered vegetation decreased by 85.1 ha/year.

Over the last 32 years, the land cover change detection between 1986 and 2018 revealed that forest, built-up areas, shrub/scattered vegetation, and water bodies increased at a rate of 510.3, 153.2, 37.2, and 9.5 ha/year, respectively. Contrary to this, the share of grassland, cropland, and bare areas diminished at a rate of 217.0, 400.8, and 92.3 ha/year, respectively (see Figure 4).

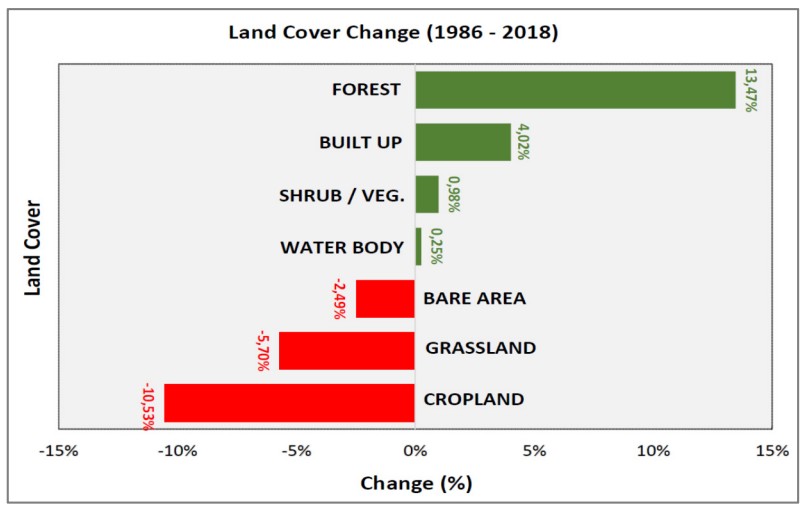

**Figure 4.** Land cover change of the study area over study period.

### 3.4. Results of Analysis of DPSIR Indicators in Relation to Land Cover Change

Figure 5 documents that 82% of the farmers perceived that the area of forests increased within the last 32 years. A total of 78% of the farmers also witnessed an increase of built-up areas, and 65% and

53% also a majority assumed that shrub/scattered vegetation and water bodies increased, respectively. A total of 85% of the farmers registered a reduction of grassland, and 79% and 77% a decrease of the area of cropland and bare areas, respectively. The farmers' perceptions of land cover changes, which were assessed by interviews, were according to the quantitative results of the land cover mapping by applying remote sensing technologies.

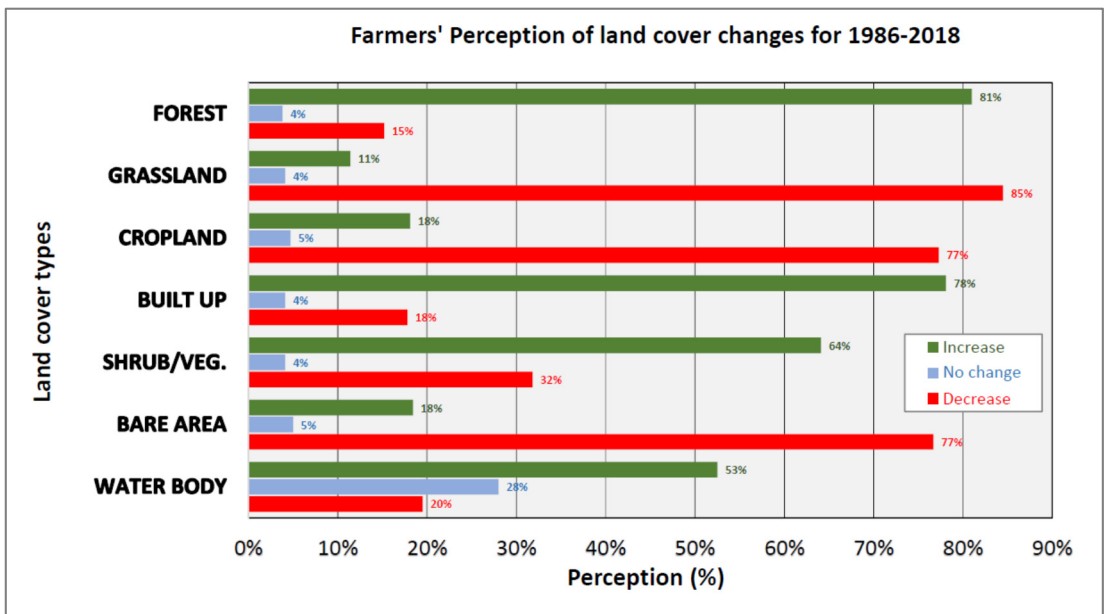

**Figure 5.** Farmers' perception of land cover changes for 1986–2018. Numbers indicate the percentage that perceived the indicated changes per land cover type (sample: 343 farmers).

### 3.4.1. Drivers of Land Cover Change

Land cover changes are the result of a bundle of driving factors. Studies have documented that drivers for land cover change are technological, economic, demographic, political, institutional, and socio-cultural factors [81].

The current study also included an investigation of drivers for land cover changes. Table 6 documents the drivers of land cover change as perceived and reported by household farmers. The main drivers in Gozamin district were increment of population growth (83.4%), land tenure system (71.4%), overuse of land (68.2%), climate change (66.8%), scarcity of grazing land (63.8%), reduced farm size (61.5%), and high wood demand (41.7%).

**Table 6.** Drivers of land cover changes as perceived by household farmers (*N* = 343 [a]).

| Drivers of Land Cover Change | Total | % |
|---|---|---|
| Increment of population growth | 286 | 83.4 |
| Overuse of land | 234 | 68.2 |
| Reduced farm size | 211 | 61.5 |
| Climate change | 229 | 66.8 |
| Rural land tenure system | 245 | 71.4 |
| High wood demand | 143 | 41.7 |
| Scarcity of grazing land | 219 | 63.8 |

[a] Total number of cases was 343 and due to a multiple response question, multiple counts are possible.

### 3.4.2. Pressures Exerted Due to Land Cover Change

The pressures exerted due to land cover change as perceived by household farmers were demand for agricultural land (80.8%), agroforestry (75.8%), over grazing of land (74.6%), over competition of communal land (70.0%), selective cutting of trees (66.2%), soil moisture change (46.1%), and increased demand for forest product (44.0%). Results are shown in Table 7.

**Table 7.** Pressures perceived by household farmers due to land cover change (*N* = 343 [a]).

| Pressures of Land Cover Change | Total | % |
|---|---|---|
| Competition on communal land | 240 | 70.0 |
| Over grazing of land | 256 | 74.6 |
| Demand for agricultural land | 277 | 80.8 |
| Increased demand for forest product | 151 | 44.0 |
| Agro-forestry | 260 | 75.8 |
| Selective cutting of trees | 227 | 66.2 |
| Soil moisture change | 158 | 46.1 |

[a] Total number of cases was 343 and due to a multiple response question, multiple counts are possible.

### 3.4.3. States of the Land due to the Land Cover Change

In the study area, the current states (conditions) observed due to land cover change by household farmers were forest cover change (85.1%), biodiversity change (78.7%), increasing land price (77.0%), land fragmentation (74.3%), rainfall variability (73.2%), and soil erosion (61.5%). All results are documented in Table 8.

**Table 8.** States perceived by household farmers due to land cover changes (*N* = 343 [a]).

| States of Land Cover Change | Total | % |
|---|---|---|
| Rainfall variability | 251 | 73.2 |
| Soil erosion | 211 | 61.5 |
| Forest cover change | 292 | 85.1 |
| Loss of soil fertility | 231 | 67.3 |
| Increase of land prices | 264 | 77.0 |
| Increased land fragmentation | 255 | 74.3 |
| Biodiversity change | 270 | 78.7 |

[a] Total number of cases was 343 and due to a multiple response question, multiple counts are possible.

### 3.4.4. Impacts of Land Cover Change

Studies have reported significant environmental and economic impacts that caused land cover changes in the highlands of Ethiopia [82–84]. For the Gozamin district, the main impacts reported by household farmers and documented in Table 9 were an increased rural to urban migration (81.3%), an increase in population (79.6%), a scarcity of land (76.1%), a decline in the productivity of land (67.9%), an increase in resource consumption (67.3%), a loss of soil quality (64.4%), and a loss of biodiversity (54.2%).

**Table 9.** Impacts of land cover changes as perceived by household farmers (*N* = 343 [a]).

| Impacts of Land Cover Change | Total | % |
|---|---|---|
| Increase rural to urban migration | 279 | 81.3 |
| Scarcity of land | 261 | 76.1 |
| Land productivity decline | 233 | 67.9 |
| Change in population size | 273 | 79.6 |
| Loss of soil quality | 221 | 64.4 |
| Loss of biodiversity | 186 | 54.2 |
| Increase resource consumption | 231 | 67.3 |

[a] Total number of cases was 343 and due to a multiple response question, multiple counts are possible.

### 3.4.5. Responses of Farmers on the Effect of Land Cover Change

The responses for household farmers' perceptions of the effects of land cover changes were raising awareness of farmers in land management (87.2%), conservation and rehabilitation of resources (82.8%), applying appropriate land use planning (68.8%), and investment in land resources (53.6%). The results are indicated in Table 10.

**Table 10.** Responses perceived by household farmers in land cover change (*N* = 343 [a]).

| Responses of Land Cover Change | Total | % |
|---|---|---|
| Conservation and rehabilitation of resource | 284 | 82.8 |
| Investment in land resource | 184 | 53.6 |
| Raising awareness of farmers in land management | 299 | 87.2 |
| Applying appropriate land use planning | 236 | 68.8 |

[a] Total number of cases was 343 and due to a multiple response question, multiple counts are possible.

### 3.4.6. DSPIR Model

Based on the answers from household farmers about their perceptions of different factor concerning the land cover change, a DSPIR model was elaborated. Figure 6 lists all the answers received from household farmers in the questionnaire survey but includes also factors mentioned in the focus group discussions.

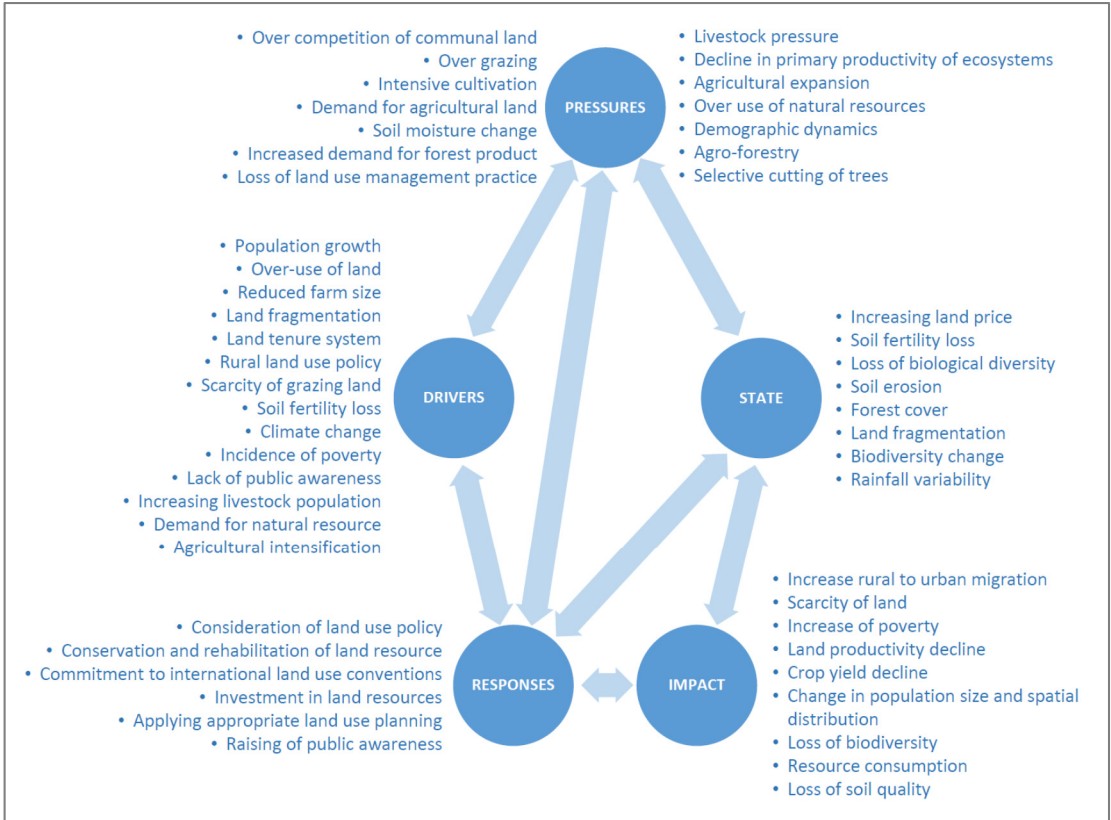

**Figure 6.** DPSIR land cover change indicators as investigated in the study area (Adapted from European Environment Agency, 2003; FAO, 2004 [80,85]).

## 4. Discussion

### 4.1. Land Cover Change

The land cover of Gozamin district indicated significant spatio-temporal changes during the study period (1986–2018) (Table 5; Figure 4). The current study identified a high rate of land cover change to forest areas (16,328.8 ha), to built-up areas (4902.5 ha), a slight annual increment rate to shrub/scattered vegetation areas (1189.4 ha), and water bodies (302.9 ha). Grassland (6944.5 ha), cropland (12,826.6 ha), and bare areas (2952.5 ha), on the contrary, showed a declining trend.

The peculiar phenomenon observed in this study was a continuous increase in forest cover area. The share of forest cover occupied 15.30%, 18.79%, and 28.77% in 1986, 2003, and 2018, respectively (Table 3). Hence, over the 32-year time period between 1986 and 2018, forest cover almost doubled. That is, from 15.30% to 28.77% (see Table 3). This result is in line with Erickson [86], who documented in Southeast Michigan a significant increase in forest cover from 8.5% to 30.6% for a specified time period.

The increase in forest area during the 1980s can be highly linked with the regeneration of natural forests, because of the afforestation program of the Derg government, the military dictatorship that governed Ethiopia from 1974 to 1991. The authors in [87] designated a strong initiative during the Derg period to preserve the remnant native trees and increase forest cover through an afforestation program in different places of Ethiopia. Another crucial reason for the expansion of areas under forest cover is attributed to the plantation of eucalyptus (*Eucalyptus globulus*). Eucalyptus has a very good performance in areas with a shortage of water and degraded soil. Almost all discussants revealed a clear preference for eucalyptus plantation than other native trees, as eucalyptus is the main source of income, fuelwood, construction materials, and means of stabilizing gullies and landslides. Therefore, directly or indirectly, eucalyptus formed the basis of the livelihood in the community.

Different studies have conveyed that there has been a continuous increase of forest cover in the highlands of Ethiopia due to eucalyptus plantations and regeneration of some natural forests [87–90]. The Food and Agriculture Organization report showed a 0.8% per year increase in forest coverage (2005–2010) in all of Ethiopia due to eucalyptus and other plantations [91]. In this study, the trend of increasing forest cover can be justified by the growth of eucalyptus. Farmers confirmed this trend with the planting of eucalyptus, but they also recognized the sustained lessening of natural (native) forests.

Local resident responses obtained from socioeconomic survey and focus group discussions confirmed the afforestation, private- and community-level tree plantation of sesbania susban, tree Lucerne, and eucalyptus trees. Especially the planting of eucalyptus tree increased the forest cover in the study area, which is in line with the growing demand for construction and fuel wood. The adaptation of eucalyptus to different agro-ecological zones of the country is resulting in increased plantation of eucalyptus by smallholders [92]. This is due to increased household and community-level tree planting, especially around homesteads, farmland, and the boundaries of farmland as used as delineation. Some household farmers changed the cultivation of their farmland from crop to trees because of economic importance.

The increase in forest coverage is linked to community-level afforestation and to reforestation practice for indigenous and multipurpose trees promoted by the government of Ethiopia in partnership with donor organizations. In addition, households are planting eucalyptus trees around homestead and farmland. As land is retired from cultivation, the land may return to forest to get cash income.

The participants in focus group discussions (FGDs), especially experts, justified the increase in forestland during the observation period of this study with the improved government strategies and policies on rural development, especially on land tenure policy. At the beginning of the period, land certification policy emerged in Ethiopia, especially in Amhara, Tigray, Oromia, and Southern Nations, Nationalities, and Peoples' (SNNP) regions. The land certification gave the farmers tenure security and belongingness of their plots. The individual household farmers planted eucalyptus trees and other vegetation resources, which gave due attention to the natural resource management. There was a good involvement of the rural community (i.e., participatory approach), non-governmental organizations and other concerned bodies for the rehabilitation (i.e., both reforestation and afforestation) of conservation of potential areas. The local community started to practice soil and water conservation activities, area closure system, as well as planting of different tree species. In order to raise seedlings for catering the plantation, both demonstrative and community nursery sites were established in different areas and millions of seedlings have been raised per annum. The government and different non-governmental organizations distributed seedlings of different tree species. Farmers intensified the planting of trees, partly converting their crop fields into eucalyptus forests, and they planted trees at the border of their plots. Farmers in the area realized the easy harvest of eucalyptus trees and the good source of income, as eucalyptus trees mature in a relatively short time.

Different studies also confirmed that although farming and settlement expanded, the trend for forest cover increased because of the incentives provided by the government for community and household-level native tree plantations. The support of the government and the effort of communities are commendable as it is yielding a positive impact in conserving the environment and the economic wellbeing of the community [93–95]. The increasing forest cover is also positively affecting the environment by reducing soil erosion and land degradation. Studies have proved that surface erosion is minimal in areas covered by vegetation [30,32,49,96,97]. Contrary to these findings, other studies documented a continuous deterioration in forest cover and a change to agricultural land in other parts of Ethiopia [83,98–101].

Built-up areas showed a significant growth during the study periods in the Gozamin district (Table 5). During the entire study period, the expansion of built-up area was mainly at the cost of cropland, and to some extent, grassland. This high exchange of area between built-up areas and cropland is attributed to the increase of rural settlements in and around the agricultural lands. Moreover, the FGD and field observation results indicated that the high demand for public institutions, like

schools and clinics, as well as for residential areas for an increasing population contributed significantly to expansion of the built-up areas. A study by Sewnet in Infraz Watershed, in the northwestern highlands of Ethiopia, documented between 1973 and 2011 an increase of areas under settlement and cropland from 4492 ha to 11,177 ha [90]. Other studies [83,102–105] also confirmed the continuous and rapid growth of built-up areas in different parts of Ethiopia caused by population growth, increase in rural settlement areas, and expansion of rural towns.

Water bodies include artificial ponds, springs, streams, and rivers. In the Gozamin district, water bodies slightly—but constantly—increased (Table 5). The explanation—as received from the farmers during the FGDs—is the water harvesting habit (pond and stream development) of local farmers for irrigation purposes. Similar to this result, a study done in Tigray, in the northern highlands of Ethiopia, indicated a permanent spring development and availability of water in the downstream areas [106]. Nevertheless, the figures about changes of water bodies are not significant, as they can be influenced by season-related water levels.

Shrub/scattered vegetation in the district increased until 2003 and decreased slightly until 2018 (Table 5). The farmers explained this fluctuation of shrub/scattered vegetation during the FGD. They reported that they intensified the planting of trees after the certification process (launched in 2002).

Cropland coverage occupied the largest share of land cover type in the study area. It held 43.21% in 1986; however, in 2003 the coverage slightly decreased and in 2018 the share of cropland shrank significantly by 10.3% (Table 5 and Figure 4). The tendency of cropland reduction in the district over the last decades, especially for the second period (2003–2018), might be because household farmers changed unproductive cropland plots to plant eucalyptus tree for long-term economic importance. In addition, the decrement of cropland is the result of population growth and the growing demand for settlements in the district. This agrees with a study done in Lenawee county, giving evidence that the number of farmlands declined 46% from 2558 ha in 1969 to 1387 ha in 1987. The decrease of farmland was mainly caused by the transformation of cropland to settlement area [86].

Overall, the area under grassland showed a persistently declining trend. During the first period, from 1986 to 2003, the grassland was reduced by 6035.1 ha (Table 5). The participants in FGDs and key informant interviews outlined that a significant part of grassland was reallocated to landless farmers and changed to cropland. However, in the second period, from 2003 to 2018, the rate of loss in grassland was only 0.75%. This trend might be highly attributed to better management of grassland by the local community and governmental bodies. The results of the FGDs documented that most of the farmers have been practicing cut-and-carry livestock feeding mechanism and area enclosures to restore the disappearing plant species and improve grass cover. In addition, the participants in FGDs confirmed that rural land was under public holding at the early stage of this study period, with the possession of rural land plots conditional upon residence in a village. This was the time of intensive government intervention and low attention was given to managing the natural resources properly. Farmers were prohibited by law to cut trees—neither for housing nor for charcoal purposes. The area was covered by different indigenous tree species. Many areas of communal land were distributed to youths for settlement or were used for grazing purpose.

Bare areas within the stated years have shown a continuously decreasing trend from 1986 to 2018 (Table 5). For the last 32 years, about 2.49% of bare area was changed into other type of land cover types (Figure 4). This is due to the availability of fixed plot of cropland in collaboration with an alarming rate of population growth negatively contributing to the decline of bare areas.

*4.2. DPSIR Indicators in Relation to Land Cover Change*

4.2.1. Drivers of Land Cover Change

The land cover change of the surface of an area is a result of combined association between demographic, socio-economic, biophysical, and institutional agents [12,100]. In this study, the farm

households identified seven factors as the main drivers for land cover change. The increment of population growth was perceived by 83.4% of the participants as a key driver for the observed land cover dynamics during the whole observation periods. The increasing population growth has led to a higher demand for land. An alarming rate of population growth results in land cover changes over time [107]. According to the current findings, studies in the northwestern highlands of Ethiopia [83,84] have indicated fast population growth as a driver for land cover change.

Another study also identified population pressure as a potential driver of land cover change. Ethiopia is one of the world's most populous countries and has a high population growth rate of 2.4% [108]. Additional studies, conducted in the northwestern highlands of Ethiopia, cited population growth as the basic driver for land cover change, especially the conversion of vegetated areas into cropland [90,93,100]. Unmet needs for family planning is the inability of women to access family planning methods, due to cultural reasons or other reasons. These unmet needs could be a reason for high fertility and thereby increasing populations. During FGDs, communities also identified population growth as the key driver of land cover change in Gozamin district.

The rural land tenure system was another prominent driver of land cover change. The participants of FGDs stated that during the 1991 government change, there was an economy close to free market economy and a privatization of government resources. As a result, farmers occupied government-owned land resources, like forests and farmlands. Participants indicated that the issuance of landholding certificates by the government, starting from 1996, has brought a sense of ownership by encouraging them to invest more in their lands. Studies confirmed that the land use certification improved tenure security [109].

In addition, overuse of land, climate change, scarcity of grazing land, and reduced farm size were also seen by the participants of the FGDs as essential factors of land cover change.

### 4.2.2. Pressures in Land Cover Change

About 81% of the participants of the questionnaire survey identified the demand for agricultural land as the key pressure for land cover change during the last decades. The increased number of people causes a higher demand on existing land resources in general, and in particular, on agricultural land, fuel wood, and construction materials. The demand for land resources creates pressure on land in the Gozamin district. The type of agriculture practiced also puts pressure on land. In this area, it is mostly rain-fed subsistence agriculture, which requires more land in order to produce more food for the growing population.

### 4.2.3. State in Land Cover Change

Household farmers identified and reported states of the land with regard to land cover changing situations. In the study area, farmers frequently mentioned forest cover change, biodiversity change, increasing land price, land fragmentation, rainfall variability, soil fertility loss, and soil erosion as current state (condition) of land due to land cover change.

### 4.2.4. Impacts of Land Cover Change

Among recognized factors, the increase in rural to urban migration was perceived by about 81% of the participants as a principal impact of land cover changes. This is due to the population increase and people, especially youth, migrate to the nearest towns because of increased prices of agricultural lands, over-consumption of irrigable water, and increased prices of food. Change in population size was the other highly perceived impact of land cover change. Another prominent impact is the scarcity of land. Studies reported that reduction in grassland has caused a lack of available suitable grazing lands, which in turn has caused over-grazing and discouraged the households from raising large-sized animals [110]. The participants also identified land productivity decline (67.9%) and loss of soil quality (64.4%) as impacts of land cover change in the area.

Previous studies reported that the land cover change through the removal of soil cover speeds up the top soil loss [82]. Almost all focus group discussants reported that the change in land cover has serious consequences for soil erosion, causes a decline in normal feed, and worsens the production of crops and livestock. In addition, the participants said that rapid population growth coupled with a decline in agricultural production (mostly caused by land cover change induced soil erosion) and unstable economic growth has posed a serious challenge of migration accompanied by social and/or political instability. The situation has, even more, affected the youth and women, who have migrated overseas in search of better livelihoods and economic opportunity. In line with this finding, the results by [83,110] identified multiple socio-economic impacts of land cover change in the northern parts of Ethiopia. The results of the FGDs realized that farmers used fertilizers on their farm plots to reverse the nutrient loss and sustain productivity of the land. About 54% of respondents perceived the negative impacts of land cover change on biodiversity losses in the study area. Studies have indicated that land cover dynamics is one of the major impacts on biodiversity persistence [111]. Also, many studies around the globe [111–117] have investigated the negative impacts of land cover change on biodiversity loss. All these studies deduced that human-made land cover change has aggravated the loss of habitats and biodiversity fragmentation by increasing the vulnerability of biological populations to speculative risk loss.

### 4.2.5. Response in Land Cover Change

Responses are understood as actions to be taken by the government to mitigate negative impacts of land cover change. The participants of the questionnaire survey as well as in the FGDs identified raising awareness of farmers in land management, conservation, and rehabilitation of resources, the application of appropriate land use planning, and the investment in land resources as the most required measures to meet the challenges of land cover changes.

## 5. Summary, Conclusions, and Recommendations

The documentation of land cover changes by means of remote sensing and the interpretation of land use changes by the DPSIR framework are proper tools for raising awareness and better understanding of drivers, pressures, states, impacts, and responses on the adverse land cover changes in a specific area. This study investigated land cover changes in the Gozamin district during a period of 32 years, from 1986 to 2018.

By classifying time series of satellite images, the land cover changes were identified, quantified, and analyzed. The accuracies achieved by the land cover classification were adequate for an overall qualitative analysis about the demands, states, pressures, impacts, and responses of land cover changes in the study area.

The land cover maps indicated a decline in cropland and grassland, with increasing trends in forest and settlement.

The DPSIR framework provides qualitative means focused on the analysis of land cover changes. It includes an explanatory platform for understanding the complexity of such change. The novelty of this study lies in combining quantitative techniques for land cover change detection, like remote sensing and ground observations, and qualitative investigations, like household surveys and focus group discussion, to capture the understanding and responses of the community to land cover change.

Population growth was identified as a key driver of land cover change, the demand for agricultural land as a key pressure. The main state (conditions) observed due to land cover change was forest cover change. As a main impact, the increase of rural to urban migration was identified by the respondents, and finally, raising awareness of farmers in land management, conservation, and rehabilitation of resources, the application of appropriate land use planning and the investment in land resources were highlighted as the key responses to all the factors mentioned above.

In general, the increase of forest land is at high rate. To maintain this trend would require an enhancement of present forest land management practices. Information about appropriate cultivation

techniques as well as about soil and water conservation measures has to be given to the farmers to mitigate land degradation and to improve the welfare of the community in the area.

The cultivation practice in the area is mainly dependent on a traditional rainfed agriculture. Livestock are fed entirely on natural grasslands. If this condition will continue in a similar manner in the future, land degradation can put the sustainability of agriculture and availability of natural resources in the area at a great risk, leading to a decline in crop production as well as to a shortage of forage for livestock. Hence, proper and integrated approaches in implementing policies and strategies related to land resources management have to be considered. Enhancing productivity using proper technologies needs to be induced to optimize the yield of crops.

Having in mind the above-mentioned aspects and the results of the current study, the following recommendations can be given to enable proper land management and conservation of remnant forest resources in the area:

- The concerned government authorities should promote yearly tree planting in collaboration with non-governmental organizations;
- Public authorities should provide incentives to the local people for protecting and restoring the native forest, as well as for guarding new plantations;
- The concerned government authorities should take appropriate steps to avoid further degradation of land and to restore the degraded lands;
- Proper land use planning should be carried out for the area prior to any developmental project being conducted in the area;
- The concerned government authorities should give attention to family planning methods to reduce the alarming population growth.

**Author Contributions:** Conceptualization, A.A.G., R.M. and T.B.; methodology, A.A.G. and T.B.; validation, A.A.G. and R.M.; formal analysis, A.A.G.; investigation, A.A.G. and T.B.; resources, A.A.G., and R.M.; data curation, A.A.G.; writing—original draft preparation, A.A.G.; writing—review and editing, R.M., C.A. and T.B.; visualization, A.A.G. and R.M.; supervision, R.M. and C.A.; project administration, R.M. and S.K.A.; funding acquisition, R.M. and S.K.A. All authors have read and agreed to the published version of the manuscript.

**Funding:** This research was enabled by a scholarship funded by the Austrian Development Cooperation within the Austrian Partnership Program in Higher Education and Research for Development (APPEAR). Project no. 113 "Implementation of Academic Land Administration Education in Ethiopia for Supporting Sustainable Development" (EduLAND2).

**Acknowledgments:** The authors thank the open access publishing was supported by BOKU Vienna Open Access Publishing Fund. In addition, the authors thank household farmers, land use and land administration experts in the study area for their collaboration during survey interviews.

**Conflicts of Interest:** The authors declare no conflict of interest. The funders had no role in the design of the study; in the collection, analyses, or interpretation of data; in the writing of the manuscript, or in the decision to publish the results.

## Appendix A

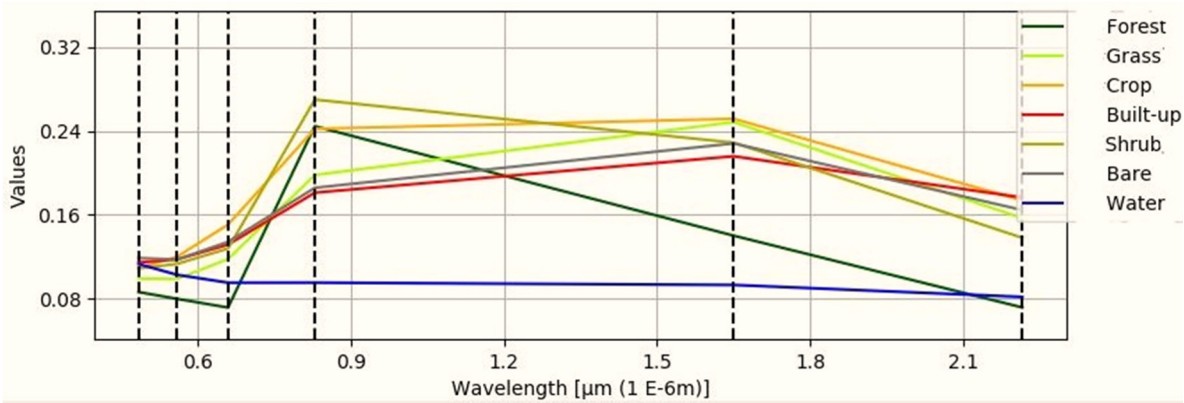

**Figure A1.** Average reflectance signatures of the seven classes_1986.

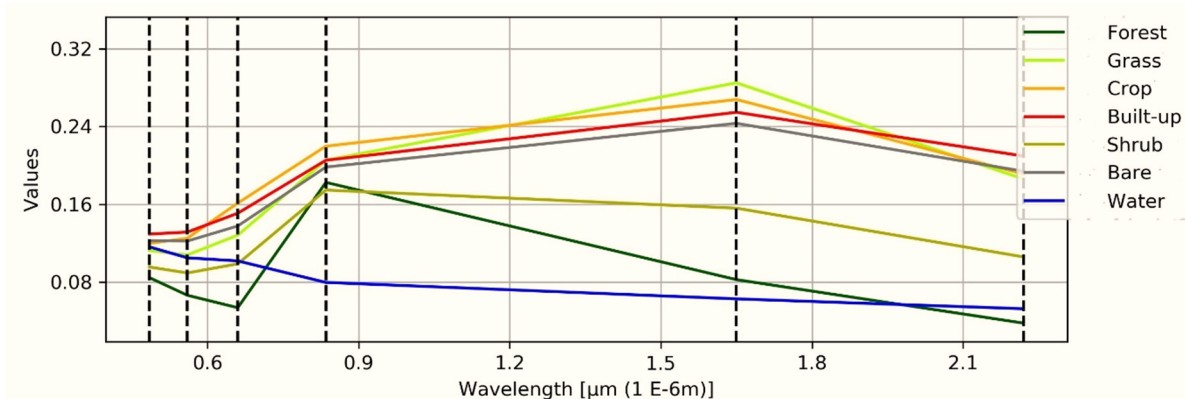

**Figure A2.** Average reflectance signatures of the seven classes_2003.

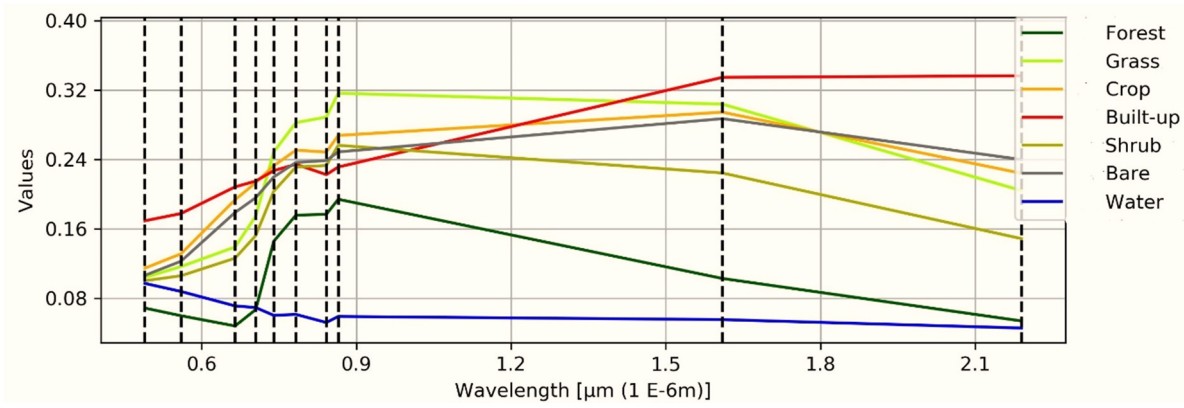

**Figure A3.** Average reflectance signatures of the seven classes_2018.

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
