# Peer review of "Analysis of Land Cover Change Detection in Gozamin District, Ethiopia: From Remote Sensing and DPSIR Perspectives"

_sustainability, doi:10.3390/su12114534_

Round 1
Reviewer 1 Report
The manuscript, entitled "Analysis of land cover change detection in Gozamin District, Ethiopia: From Remote Sensing and DPSIR perspective" by Gedefaw A.A. et al., focuses on the land cover change detection in a district area of Ethiopia over a period of 32 years (1986-2018). The detection was based on remote sensing data (Landsat-5 and -7, and Sentinel-2) and techniques (supervised classification), as well as on field work-derived data. The produced land cover maps were evaluated in terms of accuracy and constituted the base for the extraction of percentage metrics indicating the detected changes. Moreover, group discussions, interviews, and local farmers’ experiences through a household survey were applied to identify the forcing drivers for changes based on the DPSIR framework.
The manuscript deals with a topic of high interest which makes it suitable for publication in the “Sustainability” journal. The proposed methodology and its output results are certainly valuable. In general, the structure and content of the manuscript are acceptable. However, some comments and suggestions as minor revisions are provided below so that the authors take them under consideration.
Comments and suggestions
Line 32: Delete “.” Between “et” and “al”.
Introduction: Since your study focuses on the land cover detection mainly based on remote sensing data and techniques, the provided information about the relationship between remote sensing and change detection can be considered as poor (relevant mention only in lines 43-45). This part of introduction needs to be reinforced.
Lines 45-51: Due to similarity in content (referring to driving forces), these lines should be integrated in the next paragraph (lines 52-70).
Line 66: “cause” is correct.
Line 92: What does “ca.” mean?
Line 124: Either “each land cover class” or “each of land cover classes” is correct.
Lines 146-147: Rephrase this sentence.
Lines 161-163: Rephrase this sentence.
Line 170: Maybe you mean “1986 to 2018” instead of “1986 to 2003”.
Line 171: “Specific class gains and losses” is more correct.
Line 197: I think a heading like “DPSIR model for identification of drivers of land cover changes” (or something similar) is more appropriate. As you previously mentioned, land cover analysis is more related to the calculation of the relevant change percentage metrics.
Line 203: It is mentioned that “Drivers are forces that cause socio-economic changes…”, and the same time in Figure 2 it is mentioned the “Socio-economic and socio-cultural forces that changes…” as drivers. Since a confusion is caused by that, rephrase one of them.
Lines 232-233: Move “as” from “displayed as” to the location between “2018” and “land cover maps”.
Lines 391-392: Rephrase this sentence.
Lines 441-444: Rephrase this sentence.
Lines 445-447: Rephrase this sentence.
Line 450: Delete “of” after “participants”.
Line 509: Correct “Amongst” as “Among”.
Line 561: Add “of” after “increase”.
Line 564: Delete “the” after “to”.
Author Response
Response to Reviewer 1 Comments:
Dear reviewer, thanks for your precious time and for providing us with valuable feedback to improve our paper. We tried to revise our paper according to your comments and suggestions. Our detailed responses to your comments are documented below.
The manuscript, entitled "Analysis of land cover change detection in Gozamin District, Ethiopia: From Remote Sensing and DPSIR perspective" by Gedefaw A.A. et al., focuses on the land cover change detection in a district area of Ethiopia over a period of 32 years (1986-2018). The detection was based on remote sensing data (Landsat-5 and -7, and Sentinel-2) and techniques (supervised classification), as well as on-field work-derived data. The produced land cover maps were evaluated in terms of accuracy and constituted the base for the extraction of percentage metrics indicating the detected changes. Moreover, group discussions, interviews, and local farmers’ experiences through a household survey were applied to identify the forcing drivers for changes based on the DPSIR framework.
The manuscript deals with a topic of high interest which makes it suitable for publication in the “Sustainability” journal. The proposed methodology and its output results are certainly valuable. In general, the structure and content of the manuscript are acceptable. However, some comments and suggestions as minor revisions are provided below so that the authors take them under consideration.
Comments and suggestions:
Point 1: Line 32: Delete “.” Between “et” and “al”.
Response Point 1: The full stop has been deleted.
Point 2: Introduction: Since your study focuses on the land cover detection mainly based on remote sensing data and techniques, the provided information about the relationship between remote sensing and change detection can be considered as poor (relevant mention only in lines 43-45). This part of the introduction needs to be reinforced.
Response Point 2: We added relevant literature to the introduction part to emphasize the relationship between remote sensing and land cover change detection in lines 51-59 as outlined below.
Remote sensing data are proper sources for assessing land cover [10–14]. With the invention of remote sensing techniques, land cover mapping has given a useful and detailed way to improve the selection of areas designed to agricultural and urban areas of a region [15]. Remote sensing technology is also important for monitoring and quantifying the natural resources and dynamic phenomena on the Earth’s surface [16]. In recent years, remote sensing data has effectively assessed long-term changes in vegetation cover [17]. Satellite imagery is a cost-effective tool to capture and to analyze land cover data over large geographic regions. In the last decades, several techniques of land cover mapping and change detection have been developed and applied [18–26].
Point 3: Lines 45-51: Due to similarity in content (referring to driving forces), these lines should be integrated into the next paragraph (lines 52-70).
Response Point 3: Revised accordingly. The sentences that you mentioned have been integrated into the next paragraph. For your information see lines 66-84.
Point 4: Line 66: “cause” is correct.
Response Point 4: Thanks for your kind suggestions. We changed.
Point 5: Line 92: What does “ca.” mean?
Response Point 5: Thanks. We deleted ‘ca’, as it is not necessary.
Point 6: Line 124: Either “each land cover class” or “each of land cover classes” is correct.
Response Point 6: You are right, thank you for the correction.
Point 7: Lines 146-147: Rephrase this sentence.
Response Point 7: Thank you for the suggestion. The sentence has been rephrased. See line 184 ff.
Finally, this approach has a higher probability to consider minority classes that can be swamped by larger classes in unsupervised training,
Point 8: Lines 161-163: Rephrase this sentence.
Response Point 8: The sentence has been rephrased. See lines 203ff.
The accuracy of the classification was assessed using randomly selected reference sample points. The accuracy measures, such as overall accuracies, kappa coefficients, user’s and producer’s accuracies were calculated and an error matrix of the land cover classification was generated [67,68].
Point 9: Line 170: Maybe you mean “1986 to 2018” instead of “1986 to 2003”.
Response Point 9: Thank you for your advice. We corrected “1986 to 2018”.
Point 10: Line 171: “Specific class gains and losses” is more correct.
Response Point 10: Thank you for the comment. We changed the sentence (see lines 212-213).
Specific class gains and losses as well as total change and net changes of the study area were assessed [71,74].
Point 11: Line 197: I think a heading like “DPSIR model for identification of drivers of land cover changes” (or something similar) is more appropriate. As you previously mentioned, land cover analysis is more related to the calculation of the relevant change percentage metrics.
Response Point 11: Thank you for your suggestion. We changed the “DPSIR model for identification of factors of land cover changes”.
Point 12: Line 203: It is mentioned that “Drivers are forces that cause socio-economic changes…”, and at the same time in Figure 2 it is mentioned the “Socio-economic and socio-cultural forces that change…” as drivers. Since confusion is caused by that, rephrase one of them.
Response Point 12: Thanks. We rephrased line 203 to “Drivers are forces that cause socio-economic and socio-cultural forces”. For your information see line 244.
Point 13: Lines 232-233: Move “as” from “displayed as” to the location between “2018” and “land cover maps”.
Response Point 13: Thanks. We rephrased the sentence (see line 273-275)
The results of land cover classification of the Gozamin District for the years 1986, 2003, and 2018 are documented graphically in Figure 3.
Point 14: Lines 391-392: Rephrase this sentence.
Response 14: Thank you for your suggestion. The sentences have been rephrased. See lines 435-438.
The land certification gave the farmers tenure security and belongingness of their plots. The individual household farmer planted eucalyptus trees and other vegetation resources, which gave due attention to natural resource management.
Point 15: Lines 441-444: Rephrase this sentence and
Point 16: Lines 445-447: Rephrase this sentence.
Response Point 15 and Point 16: Thanks for comment. The sentences have been rephrased. See lines 485-496.
The tendency of cropland reduction in the district over the last decades, especially for the second period (2003-2018), might be household farmers changed unproductive cropland plots to plant eucalyptus tree for long-term economic importance. In addition, the decrement of cropland is the result of population growth and the growing demand for settlements in the district. This agrees with a study done in Lenawee county, giving evidence that the number of farmlands declined 46% from 2558 ha in 1969 to 1387 ha in 1987. The decrease of farmland mainly was caused by the transformation of cropland to the settlement area [87].
Point 17: Line 450: Delete “of” after “participants”.
Response Point 17: The word was deleted.
Point 18: Line 509: Correct “Amongst” as “Among”.
Response Point 18: Both words would be corrected. Nevertheless, we appreciate your suggestion and will use “Among”.
Point 19: Line 561: Add “of” after “increase”.
Response Point 19: Thanks for comment. We added “of” after “increase”.
Point 20: Line 564: Delete “the” after “to”.
Response Point 20: Thanks for the advice. We deleted the word “the”.
Reviewer 2 Report
General comments:
Manuscript is interesting and demonstrates the topical application of the DPSIR framework in Ethiopian setting. There are some issues with this manuscript, mainly related to the readability and composition of the manuscript.
Please review the quality of your English throughout the manuscript.
I would like to see some more specific findings of this study reported at the end of the abstract.
Specific comments:
L39 – “Lately, studies on land cover change detection have drawn devotion of numerous researchers [6,7].” – please rephrase
L43 – “the Earth” is a proper noun
L56 – change “the combination” to “a combination”
L74 – change “increased” to ”increasing”
L93 – “(m.a.s.l)” is not needed if you’ve already spelled it out and use only once
L138-139 – Why it was necessary to check the reliability of reference data in group discussions and interviews, if you collected reference data in the field?
L146-147 – please rephrase the sentence
L153-155 – specify what was the size of the majority filter window
L164-167 – I suggest to remove these two contradictory sentences
L177-178 – please explain what “landholder household farmers” are. I suggest rephrasing
L258 – I suggest bolding the overall changes (from 1986 to 2018) to make them more distinguishable from the rest of results
L277-279 – rephrase the first part of this sentence
L295 – change “of bundle” to “of a bundle”
L353-354 – “This is in line with Erickson [86] documenting in Southeast Michigan a significant increase in forest cover from 8.5% to 30.6%.” – I don’t see how this is related to your results stated in this paragraph.
L344 – Please shorten 4.1. section, most of the reporting of results should be moved to Results
L589-592 – Remove this paragraph, it tells nothing
Author Response
Response to Reviewer 2 Comments:
Dear reviewer, thanks for your precious time and for providing us with valuable feedback to improve our paper. We tried to revise our paper according to your comments and suggestions. Our detailed responses to your comments are documented below.
General comments:
The manuscript is interesting and demonstrates the topical application of the DPSIR framework in the Ethiopian setting. There are some issues with this manuscript, mainly related to the readability and composition of the manuscript.
Point 1: Please review the quality of your English throughout the manuscript.
Response Point 1: We reviewed the quality of English accordingly and rephrased some text.
Point 2: I would like to see some more specific findings of this study reported at the end of the abstract.
Response Point 2: Thank you for your suggestions. The authors have added more specific findings at the end of the abstract. We added specific findings as following in lines 22-31.
Thus, quantitatively land cover change detection between 1986 and 2018 revealed that cropland, grassland, and bare areas declined by 10.53%, 5.7%, and 2.49%. Forest, built-up, shrub/scattered vegetation and water bodies expanded by 13.47%, 4.02%, 0.98% and 0.25%. Household surveys and FGDs identified the population growth, the rural land tenure system, the overuse of land, the climate change, and the scarcity of grazing land as drivers of these land cover changes. Major impacts were rural to urban migration, population size change, scarcity of land, and decline in land productivity. The outputs from this study could be used to assure sustainability in resource utilization, proper land use planning, and proper decision making by the concerned government authorities.
Point 3: L39 – “Lately, studies on land cover change detection have drawn devotion of numerous researchers [6,7].” – please rephrase.
Response Point 3: Thank you for your suggestions. The sentence was rephrased. See line 46-48.
Recently, studies to detect changes in land cover have attracted the attention of numerous researchers
Point 4: L49 – “the Earth” is a proper noun
Response Point 4: Thank you. We corrected as “the Earth”.
Point 5: L56 – change “the combination” to “a combination”
Response Point 5: Thanks for your suggestion. We changed “the” to “a”.
Point 6: L74 – change “increased” to ”increasing”
Response Point 6: We adapted the text according to your suggestion.
Point 7: L93 – “(m.a.s.l)” is not needed if you’ve already spelled it out and use only once
Response Point 7: We deleted “m.a.s.l”. See line 128.
Point 8: L138-139 – Why it was necessary to check the reliability of reference data in group discussions and interviews if you collected reference data in the field?
Response Point 8: Thank you for your comment, which gave evidence about an imprecise wording. We tried to clarify this sentence (lines 174-177)
With regard to an existing time lag between the acquisition date of satellite images and the assessment of reference data (Google Earth data and field survey), the reliability of reference data was checked in group discussions and interviews with farmers.
Point 9: L146-147 – please rephrase the sentence
Response Point 9: Thank you for the suggestion. The sentence has been rephrased. See line 184ff.
Finally, this approach has a higher probability to consider minority classes that can be swamped by larger classes in unsupervised training.
Point 10: L153-155 – specify what was the size of the majority filter window
Response Point 10: The size of the major filter window is 7 by 7 kernel size. The information was added to the text. See line 194-196.
The size of the majority filter window was chosen with 7 by 7 as larger kernel sizes would produce a higher smoothing in the classification image
Point 11: L164-167 – I suggest to remove these two contradictory sentences
Response Point 11: We agree with You and delete both sentences. We only wanted to justify, that we used a standard classification algorithm and did not spend too much effort to improve the accuracy of the classification. Nevertheless, the achieved results have sufficient accuracy.
Point 12: L177-178 – please explain what “landholder household farmers” are. I suggest rephrasing
Response Point 12: Thank you for your comment. This was a mistake. We replaced ‘landholder household farmers’ by ‘household farmers’ throughout the manuscript.
Point 13: L258 – I suggest bolding the overall changes (from 1986 to 2018) to make them more distinguishable from the rest of the results
Response Point 13: Thank you. We changed Table 5 to make bold the results showing “1986 to 2018” to distinguish from the rest of the results. See line 299ff.
Point 14: L277-279 – rephrase the first part of this sentence
Response 14: We rephrased the sentences you mentioned. See lines 318ff.
Over the last 32-years, the land cover change detection between 1986 and 2018 revealed that forest, built-up areas, shrub/scattered vegetation, and water bodies have increased with a rate of 510.3, 153.2, 37.2 and 9.5 ha/year respectively.
Point 15: L295 – change “of bundle” to “of a bundle”
Response Point 15: We added the article in the sentence.
Point 16: L353-354 – “This is in line with Erickson [86] documenting in Southeast Michigan a significant increase in forest cover from 8.5% to 30.6%.” – I don’t see how this is related to your results stated in this paragraph.
Response Point 16: The result of our study gives evidence that forest cover increased from 15.30% to 28.77% over the 32 years time period “1986 to 2018”. Erickson’s result also documents an increase in forest cover increases by 8.5% to 30.6% for a specified time period.
Point 17: L344 – Please shorten 4.1. section, most of the reporting of results should be moved to Results
Response 17: Thank you for your suggestion. We shortened the text by summarizing the trends and by giving reference to details in chapter 3.
Point 18: L589-592 – Remove this paragraph, it tells nothing
Response Point 18: Thank you. We deleted your mentioned paragraph.